# Efficacy of Intravesical Botulinum Toxin A Injection in the Treatment of Refractory Overactive Bladder in Children

**DOI:** 10.3390/jpm13040616

**Published:** 2023-03-31

**Authors:** Yu-Hua Fan, Hann-Chorng Kuo

**Affiliations:** 1Department of Urology, Taipei Veterans General Hospital, Taipei 112, Taiwan; yhfan2@gmail.com; 2Department of Urology, College of Medicine, National Yang Ming Chiao Tung University, Taipei 112, Taiwan; 3Department of Urology, Hualien Tzu Chi Hospital, Buddhist Tzu Chi Medical Foundation and Tzu Chi University, Hualien 970, Taiwan

**Keywords:** overactive bladder, botulinum toxin A, urodynamic study

## Abstract

This study aimed to evaluate the efficacy of intravesical botulinum toxin A (BoNT-A) injections for the treatment of pediatric overactive bladder (OAB) by exploring the differential treatment outcomes in children with different OAB etiologies and those who received additional intrasphincteric BoNT-A injections. We performed a retrospective review of all pediatric patients who received intravesical BoNT-A injections between January 2002 and December 2021. All patients underwent a urodynamic study at baseline and three months after BoNT-A administration. A Global Response Assessment (GRA) score of ≥2 at three months after BoNT-A injection was defined as successful treatment. Fifteen pediatric patients (median age, 11 years), including six boys and nine girls, were enrolled in the study. A statistically significant decrease in detrusor pressure from baseline to three months postoperatively was observed. Thirteen (86.7%) patients reported successful results (GRA ≥ 2). The cause of OAB and additional intrasphincteric BoNT-A injections did not affect the improvement in urodynamic parameters and treatment success. The study demonstrated that intravesical BoNT-A injection is effective and safe for the treatment of neurogenic and non-neurogenic OAB in children refractory to conventional therapies. Additionally, intrasphincteric BoNT-A injections do not provide additional benefits in the treatment of pediatric OAB.

## 1. Introduction

Overactive bladder (OAB) is a syndrome characterized by urinary urgency/frequency, with or without urinary incontinence, in the absence of a causative infection or pathological conditions [1]. When accompanied by incontinence, OAB can be particularly challenging for pediatric patients and their families, as it can have adverse effects on children’s development and self-esteem [2]. This condition poses a significant impact on daily activities and quality of life [3]. OAB is observed globally, and the pediatric population experiences a high prevalence of OAB, with a prevalence of 5–12% in children (5–10 years of age) and 0.5% in older adolescents (16–18 years of age) [3]. OAB can be further categorized into neurogenic and non-neurogenic OAB. The most common etiology of neurogenic OAB in children is spina bifida, which is responsible for up to 93% of cases. Sacral agenesis, imperforate anus, and lesions of the spinal cord are each responsible for 1% of cases, and cerebral palsy is responsible for 3%. Other less frequent causes are cerebral/spinal tumors and pelvic surgery [4]. Neurogenic bladder may result in decreased bladder capacity, low compliance, and high intravesical pressure [5]. Children with neurogenic bladder are at an increased risk of urinary tract infection and renal function deterioration [6]. Non-neurogenic OAB may arise from various etiologies, including anatomic, inflammatory, and idiopathic causes. Chung et al. conducted a prospective multicenter study to investigate the risk factors for non-neurogenic OAB in Korean children (5–13 years of age) [7]. Enuresis, constipation, fecal incontinence, urinary tract infection, and delayed toilet training were considered risk factors associated with non-neurogenic OAB.

The primary treatment of non-neurogenic OAB is urotherapy, with bladder re-education or rehabilitation, mostly accompanied by antispasmodic agents [8]. Routine management of neurogenic OAB includes oral medications such as anticholinergics or beta-3 agonists, timely clean intermittent catheterization, or long-term indwelling catheterization [9]. The goal is to reduce intravesical pressure to protect renal function. Botulinum toxin A (BoNT-A) is a possible treatment for children with non-neurogenic OAB unresponsive to a specific urotherapy and oral medications; in addition, it is a possible treatment for children with neurogenic OAB refractory to clean intermittent catheterization and medications. Produced by the Gram-negative, rod-shaped, anaerobic bacterium Clostridium Botulinum, BoNT-A is a potent neurotoxin that works by inhibiting the calcium-mediated release of acetylcholine vesicles at the presynaptic neuromuscular junction in peripheral nerve endings. This mechanism leads to a temporary flaccid muscle paralysis, which is the basis of its efficacy [10]. Although BoNT-A is not licensed for use in children with neurogenic or non-neurogenic OAB, it has been widely used to treat pediatric OAB refractory to conventional therapy [11]. Most studies report using onabotulinum toxin (100–500 IU, 10–50 IU/kg according to body weight), and the number of injection sites ranged from 20 to 50 [6].

However, there is little evidence on the efficacy of BoNT-A in the treatment of pediatric OAB, especially non-neurogenic OAB. Furthermore, few studies have compared the treatment outcomes of intravesical BoNT-A between neurogenic and non-neurogenic OAB and evaluated the benefit of additional urethral sphincter BoNT-A on bladder storage function. Therefore, this study aimed to evaluate the efficacy of intravesical BoNT-A injections for the treatment of pediatric OAB, exploring the differential treatment outcomes in children with different OAB etiologies and those who received additional intrasphincteric BoNT-A injections.

## 2. Materials and Methods

### 2.1. Study Design and Subjects

Medical records of consecutive pediatric patients aged up to 18 years who received a first dose of intravesical BoNT-A injection between January 2002 and December 2021 were retrospectively reviewed. The participants may have received more than one dose of BoNT-A, but we only investigated the effect of the first BoNT-A injection. All study participants’ parents or guardians provided informed consent. The Ethics Committee of Hualien Tzu Chi Hospital, Buddhist Tzu Chi Medical Foundation, approved this study (IRB110-147-B), and all methods were performed in accordance with the principles stated in the Declaration of Helsinki [12].

### 2.2. Urodynamic Assessment

Urodynamic studies have become a major tool in evaluating lower urinary tract dysfunction in children. The urodynamic study in conjunction with fluoroscopy records fluoroscopic images during testing that offer several benefits. The shape of the bladder and bladder neck during filling and voiding, appearance of the urethra during voiding, the volume and pressure when vesico-ureteral reflux occurs, and the effect of voiding on vesico-ureteral reflux can be objectively noted [13]. Therefore, all patients underwent videourodynamic studies (VUDSs) before the administration of the first dose of intravesical BoNT-A injection. VUDS were repeated three months after the first injection. Patients with no recorded VUDSs after the first injection were excluded from analysis. VUDSs were performed according to International Children’s Continence Society (ICCS) guidelines [13].

### 2.3. BoNT-A Injection

All the participants received intravesical BoNT-A injection according to the standard of care at our institution. All the injections were performed by a single surgeon (HCK). Intravesical BoNT-A injection is an inpatient treatment option performed under intravenous anesthesia that involves 20 trigone-sparing injections utilizing a flexible cystoscope. Prophylactic antibiotics are administered at the time of intravesical BoNT-A injection and continued for one day. The dose of BoNT-A (onabotulinum toxin A, Botox^®^) for the bladder and urethra in children is 5–10 U/kg, depending on the severity of the disease. The total dose of BoNT-A did not exceed 200 U/kg per treatment, including detrusor and urethral sphincter injections. An additional BoNT-A injection into the urethral sphincter was performed for concurrent dysfunctional voiding or detrusor sphincter dyssynergia. Because previous medications failed to treat lower urinary tract symptoms before BoNT-A injection, no additional pharmacological or non-pharmacological intervention was given after BoNT-A treatment.

### 2.4. Outcome Measures

#### 2.4.1. Primary Outcome

Primary outcome measures were differences in VUDS parameter values regarding bladder storage function before and after the first dose of intravesical BoNT-A injection. Bladder capacity varies with age. A known formula ([age in years + 1] × 30) exists to determine the satisfactory bladder capacity for age in children [1]. Therefore, cystometric capacity was recorded as the ratio of measured (bladder volume at the end of filling cystometry) to expected bladder capacity ([age in years + 1] × 30) in the present study. All definitions of VUDS parameters complied with ICCS glossary definitions, unless otherwise specified [13].

#### 2.4.2. Secondary Outcome

The secondary outcome measure was the effectiveness of intravesical BoNT-A injection, as determined by the Global Response Assessment (GRA) measured three months after the first dose of BoNT-A injection. GRA is commonly used to assess the treatment outcome of functional urology such as lower urinary tract symptoms, OAB, and voiding dysfunction. The symptoms might improve or exacerbate to different degrees after treatment. The children or their parents were asked to rate overall changes in bladder symptoms. GRA, categorized as −3, −2, −1, 0, 1, 2, and 3, indicated markedly worse, moderately worse, mildly worse, no change, mildly improved, moderately improved, and markedly improved bladder symptoms, respectively. A GRA score ≥2 was defined as successful treatment; otherwise, the treatment was considered to have failed [14].

### 2.5. Statistical Analysis

Data are presented as the medians (range). A paired *t*-test was used to compare VUDS parameter values before and after the first dose of intravesical BoNT-A injection. To compare the effects of intravesical BoNT-A injection in patients with and without relevant neurological disease, we categorized the patients into two groups: neurogenic OAB and non-neurogenic OAB. Neurogenic OAB was defined as OAB that occurred as a result of an injury or disorder of any part of the nervous system including spinal cord injury, spinal dysraphism, or intracranial lesions [15]. Furthermore, we grouped the patients into bladder BoNT-A injection only and bladder plus urethral sphincter BoNT-A injection groups to evaluate the effect of additional urethral sphincter BoNT-A injections on the change in VUDS parameter values. The Mann–Whitney U test was performed to compare the change in VUDS parameter values between the two groups. Statistical significance was set at *p* < 0.05. All statistical tests were performed using IBM SPSS Statistics for Macintosh, ver. 21 (IBM Corp., Armonk, NY, USA).

## 3. Results

Nineteen patients were initially screened, and four patients were excluded because there was no recorded VUDS after the first injection of BoNT-A. Fifteen pediatric patients, including six boys and nine girls, were enrolled in the study. The median age of the participants was 11 years (range, 3–17 years). Eleven (73.3%) patients had neurogenic OAB alongside a relevant neurological disease: nine with myelomeningocele and two with spinal cord injury. In two patients with spinal cord injury, one had been diagnosed with a T7-11 spinal cord injury with an American spinal injury association Impairment Scale (AIS) grade A one year prior to the first dose of BoNT-A, and the other one had been diagnosed with a C4-5 spinal cord injury with AIS grade A two years before the first dose of BoNT-A. Four (26.7%) patients received an additional BoNT-A injection into the urethral sphincter for concurrent dysfunctional voiding (n = 2) or detrusor sphincter dyssynergia (n = 2).

### 3.1. Urodynamic Outcomes

A statistically significant decrease in the maximum detrusor pressure from baseline (median, 25 cmH_2_O; range, 0–80 cmH_2_O) to three months (median, 15 cmH_2_O; range, 0–70 cmH_2_O) was observed (*p* = 0.013, Table 1). Other urodynamic parameters of bladder storage function, including bladder compliance and the ratio of measured-to-expected bladder capacity, did not have significant changes after intravesical BoNT-A injection.

#### 3.1.1. Neurogenic OAB vs. Non-Neurogenic OAB

Maximum detrusor pressure decreased significantly only in the neurogenic OAB group. In the neurogenic OAB group, the pre-BoNT-A median maximum detrusor pressure was 27 cmH_2_O (range, 0–61 cmH_2_O), which decreased to 15 cmH_2_O (range, 0–41 cmH_2_O) after BoNT-A was administered (*p* = 0.040, Table 2). Bladder compliance and the ratio of measured-to-expected bladder capacity did not improve significantly after the administration of BoNT-A in either group. Changes in maximum detrusor pressure, bladder compliance, and the ratio of measured-to-expected bladder capacity after BoNT-A administration were not significantly different between the two groups (Figure 1).

#### 3.1.2. Bladder BoNT-A Only vs. Bladder plus Urethral Sphincter BoNT-A

The maximum detrusor pressure decreased significantly in the bladder BoNT-A only group. In the bladder BoNT-A only group, the pre-BoNT-A median maximum detrusor pressure was 20 cmH_2_O (range, 0–61 cmH_2_O) and decreased to 13 cmH_2_O (range, 0–41 cmH_2_O) after BoNT-A was administered (*p* = 0.009, Table 2). Bladder compliance and the ratio of measured-to-expected bladder capacity did not improve significantly after BoNT-A administration in either group. Changes in maximum detrusor pressure, bladder compliance, and the ratio of measured-to-expected bladder capacity after BoNT-A were not significantly different between the two groups (Figure 2).

### 3.2. Subjective Therapeutic Outcome

Thirteen (86.7%) patients reported successful outcomes (GRA ≥2). Subgroup analyses revealed that nine (81.8%) patients with neurogenic OAB and four (100.0%) patients with non-neurogenic OAB reported successful outcomes. Similarly, nine (81.8%) patients who received bladder BoNT-A injection only and four (100.0%) patients who received bladder BoNT-A injection plus urethral sphincter BoNT-A injection reported successful outcomes. The cause of OAB and an additional intrasphincteric BoNT-A injection did not affect patient satisfaction with the treatment (*p* = 0.376). No treatment-related adverse events, including urinary tract infection and de novo intermittent self-catheterization, were observed.

## 4. Discussion

The present study demonstrated that intravesical BoNT-A injections significantly improved the maximum detrusor pressure in pediatric patients with OAB. Nevertheless, subgroup analyses revealed that the maximum detrusor pressure decreased significantly only in the neurogenic OAB and bladder BoNT-A only groups. We observed no significant changes in other urodynamic parameters regarding bladder storage function, including bladder compliance and the ratio of measured-to-expected bladder capacity after intravesical BoNT-A injection. Furthermore, the cause of OAB and an additional intrasphincteric BoNT-A injection did not affect the change in VUDS parameter values. Although the study participants merely exhibited a significant improvement in maximum detrusor pressure after intravesical BoNT-A injection, successful outcomes were subjectively reported in over 80% of the patients.

Urodynamic changes were reported as outcome measures in most studies involving the use of intravesical BoNT-A injections in children with neurogenic bladders. Although the items emphasized in each study were different, the studies mainly focused on detrusor pressure, bladder capacity, and bladder compliance. Kim et al. evaluated 37 children with neurogenic detrusor overactivity who received their first BoNT-A intradetrusor injection [16]. A significant decrease in detrusor pressure and cystometric capacity at one month after treatment was observed. However, there was no significant improvement in bladder compliance. Hascoet et al. assessed the effectiveness of intradetrusor BoNT-A injections in 53 children with spina bifida [17]. Cystometric capacity and bladder compliance significantly improved after the first BoNT-A injection, but maximum detrusor pressure did not decrease significantly. Kajbafzadeh et al. evaluated the efficacy of BoNT-A in the treatment of 26 children with neurogenic OAB caused by myelomeningocele [18]. Four months after the procedure, maximal detrusor pressure and bladder capacity improved significantly. Our study demonstrated that intravesical BoNT-A injections significantly improved the maximum detrusor pressure in pediatric patients with OAB. However, we observed no significant changes in bladder compliance and capacity. Variable urodynamic changes after BoNT-A injections might be attributed to baseline urodynamic characteristics, heterogeneous etiologies of OAB, and BoNT-A dosage.

Studies on BoNT-A in children with non-neurogenic OAB are scarce, and objective urodynamic parameters are lacking. Marte et al. and Hoebeke et al. determined the results of 21 and 15 children, respectively, with refractory non-neurogenic OAB who underwent intradetrusor BoNT-A injections. Both reported significant improvement in the incontinence episodes after the treatment, but neither included objective urodynamic assessment [19,20]. Al Edwan et al. prospectively enrolled 46 children with idiopathic OAB receiving intravesical BoNT-A injections [21]. Significant improvements in maximum detrusor pressure, compliance, and bladder capacity from baseline to three months were demonstrated. Furthermore, 54% of the patients described their condition as greatly improved and 30% as improved. Our study enrolled only four children with non-neurogenic OAB and reported no significant improvements in maximum detrusor pressure, bladder compliance, or bladder capacity after bladder BoNT-A injections. However, all patients reported successful outcomes (GRA score ≥ 2).

In addition to improvements in detrusor–sphincter dyssynergia and post-voiding residual urine volume, BoNT-A injections in the external urethral sphincter have been demonstrated to decrease maximum detrusor pressure [22,23]. Nevertheless, few studies have evaluated the additional effect of concomitant intrasphincteric BoNT-A injections on bladder storage function in patients with OAB who received intravesical BoNT-A injections. Safari et al. compared the urodynamic results between intravesical BoNT-A injections with and without intrasphincteric BoNT-A injections in treating bladder hyperreflexia in children with myelomeningocele, revealing no extra benefit for injections in both detrusor and external urethral sphincter, with regard to maximum detrusor pressure and bladder capacity [24]. Nonetheless, BoNT-A injections in both the sphincter and detrusor significantly improved voiding functions, such as decreasing post-voiding residual volume and prevalence of detrusor sphincter dyssynergia compared with the results obtained with the use of intradetrusal injections alone. Our results are similar to those of previous studies. Pediatric patients who received injections in both the bladder and external urethral sphincter did not show significant improvement in bladder compliance, maximal detrusor pressure, or bladder capacity compared with the results obtained with the use of bladder injections alone. BoNT-A injections in both the sphincter and bladder did not appear to have an extra benefit on bladder storage function compared to the use of bladder injections alone.

The patient’s reported response was also an important outcome measure. Deshpande et al. evaluated the subjective outcomes of intradetrusor BoNT-A injections in seven pediatric patients with spina bifida using validated questionnaires regarding the severity (continence score) and perception (satisfaction score) of incontinence [25]. The continence score improved significantly one month after BoNT-A injection; however, there was no significant change in the satisfaction score. Kim et al. assessed subjective outcomes using the Patient Global Impression of Improvement (PGI-I) at three months after bladder BoNT-A injection in children with neurogenic detrusor overactivity [16]. A total of 54.1% (20/37) of participants were categorized as responders. Our study evaluated patient satisfaction using GRA measured at three months after BoNT-A injection and reported successful outcomes in over 80% of the study participants with OAB, regardless of the cause of OAB and an additional intrasphincteric BoNT-A injection. Variable patient satisfaction might be attributed to the timing of assessment, heterogeneous etiologies of OAB, dosage of BoNT-A, and outcome measures.

The discrepancy between the improvement in urodynamic parameters and patients’ subjective satisfaction with BoNT-A intradetrusor injection has been previously reported [26]. In the present study, non-neurogenic OAB and bladder plus urethra BoNT-A groups had no significant change in any urodynamic parameters regarding bladder storage function after intravesical BoNT-A injection. Nevertheless, all study participants in both groups reported successful outcomes. This finding would support the routine use of urodynamics in the follow-up of children with non-neurogenic OAB because the patient’s perception in this population might not be reliable. However, both the non-neurogenic OAB and bladder plus urethra sphincter BoNT-A groups included only four children. A prospective study with more cases is, therefore, needed to explore the discrepancy between clinical and urodynamic outcomes. Furthermore, the lack of control group in the present study raises the concern that the placebo effect might account for the disconcordance between subjective and objective outcome measures. Placebo effects are particularly strong in subjective outcomes (e.g., pain) [27], functional conditions (e.g., irritable bowel syndrome) [28], and fatigue [29]. While placebo effects have a greater impact on subjective versus objective outcome measures, they appear to have a significant influence on both in OAB [30]. This implies that the placebo has a statistically significant effect on improving symptoms and signs associated with OAB. Ultimately, this needs to be evaluated in larger case-controlled studies.

The present study did not report any treatment-related adverse events. According to the latest systematic review of BoNT-A injections in children with neurogenic bladder conducted by Wu et al., 6 of 16 retrieved studies mentioned adverse effects of BoNT-A injections, and the most common adverse effect was temporary hematuria (10.9–80.9%) [6]. Post-injection urinary tract infection were the other reported adverse effects (2.1–14.3%) [6]. BoNT-A injections appeared to be safe, as only a few patients had adverse events.

The major drawbacks of the study are its relatively small sample size, uneven distribution between the subgroups, and the retrospective nature, which might prevent significant results and identification of predictive factors for better treatment outcomes. Additionally, all the participants were Taiwanese and recruited only from Hualien Tzu Chi Hospital. The results may not be generalizable outside of this context for other ethnic groups and clinics administering BoNT-A treatment where the procedures of BoNT-A injections might be different. Nonetheless, our study investigated the treatment outcomes in children with OAB refractory to conventional therapy. The pediatric patients should have a urodynamic study performed both at baseline and after BoNT-A treatment and thus, the case number is usually not many. Additionally, we performed a subgroup analysis to explore potential predictive factors for treatment outcomes, even accounting for the imbalance in the size of the subgroups. Studies investigating children with non-neurogenic OAB receiving bladder BoNT-A and children with OAB that underwent bladder plus urethral sphincter BoNT-A are uncommon. The results are valuable and might provide a viable treatment option for future clinical practice.

The absence of a frequency–volume chart, which is an objective tool for measuring changes in storage symptoms, was also a limitation of this study. The frequency–volume chart records objective information regarding number of voids, their distribution (day and night), voided volumes, and episodes of urgency and/or leakage. However, several studies have reported that changes in storage symptoms are congruent with general response assessments [31,32].

## 5. Conclusions

Although maximum detrusor pressure decreased significantly only in children with neurogenic OAB who received intravesical BoNT-A injections, successful outcomes were subjectively reported in 80% of children with neurogenic OAB and those with non-neurogenic OAB. No treatment-related adverse events were observed. Concomitant injections of BoNT-A into the external urethral sphincter did not provide additional benefits for OAB. Due to several limitations, including small sample size and the lack of control group, the results of the present study should be cautiously interpreted. Larger case-controlled studies are needed.

## Figures and Tables

**Figure 1 jpm-13-00616-f001:**
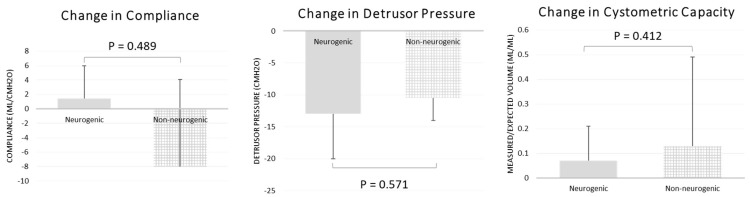
Change in bladder compliance, detrusor pressure, and cystometric capacity, comparing neurogenic OAB group with non-neurogenic OAB group after BoNT-A injection. There was no significant improvement in bladder compliance, detrusor pressure, or cystometric capacity between groups.

**Figure 2 jpm-13-00616-f002:**
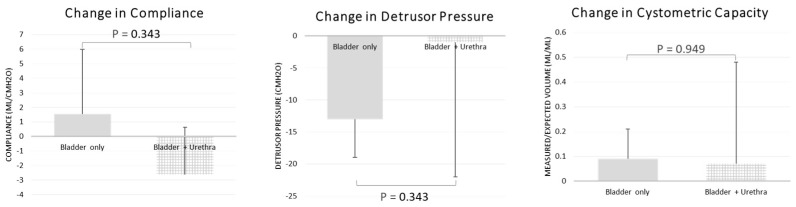
Change in bladder compliance, detrusor pressure, and cystometric capacity, comparing bladder BoNT-A group with bladder plus urethra BoNT-A group after BoNT-A injection. There was no significant improvement in compliance, detrusor pressure, or cystometric capacity between groups.

**Table 1 jpm-13-00616-t001:** Change in urodynamic parameters from before to after BoNT-A injection (n = 15).

	Median (Range)		
	Before BoNT-A	After BoNT-A	*p* Value
Compliance (mL/cmH_2_O)	8.15 (2.23–340.00)	11.54 (0.88–75.00)	0.820
Measured/expected CBC (mL/mL)	0.37 (0.05–1.15)	0.52 (0.21–1.09)	0.215
Detrusor pressure (cmH_2_O)	25 (0–80)	15 (0–70)	0.013
Post-void residual (mL)	100 (0–250)	100 (0–300)	0.780

Measured/expected CBC, the ratio of measured-to-expected cystometric bladder capacity; BoNT-A, botulinum toxin A.

**Table 2 jpm-13-00616-t002:** Subgroup analysis of the change in urodynamic parameters from before to after BoNT-A injection.

	Neurogenic OAB	Non-Neurogenic OAB	Bladder BoNT-A	Bladder + Urethra BoNT-A
No. pts	11	4	11	4
Compliance (mL/cmH_2_O)				
Before BoNT-A	8.15 (2.23–340.00)	18.24 (2.40–180.00)	8.15 (3.08–340.00)	15.77 (2.23–30.00)
After BoNT-A	10.42 (0.88–70.75)	12.33 (3.27–75.00)	11.54 (0.88–75.00)	7.71 (2.11–24.86)
*p* Value	0.461	0.343	0.316	0.283
Measured/expected CBC (mL/mL)				
Before BoNT-A	0.54 (0.05–1.15)	0.31 (0.22–0.60)	0.37 (0.05–1.15)	0.53 (0.22–0.94)
After BoNT-A	0.49 (0.25–1.09)	0.61 (0.21–0.81)	0.52 (0.21–1.09)	0.60 (0.34–0.81)
*p* Value	0.694	0.238	0.215	0.903
Detrusor pressure (cmH_2_O)				
Before BoNT-A	27 (0–61)	23 (12–80)	20 (0–61)	33 (5–80)
After BoNT-A	15 (0–41)	19 (1–70)	13 (0–41)	27 (15–70)
*p* Value	0.04	0.265	0.009	0.771
Post-void residual (mL)				
Before BoNT-A	150 (0–250)	10 (0–210)	150 (0–250)	60 (0–150)
After BoNT-A	100 (0–300)	100 (0–300)	130 (0–300)	70 (0–200)
*p* value	0.598	0.215	0.954	0.774

Data are presented as median (range). OAB, overactive bladder; Measured/expected CBC, the ratio of measured-to-expected cystometric bladder capacity; BoNT-A, botulinum toxin A.

## Data Availability

All data generated or analyzed during this study are included in this published article.

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
