# Peer review of "Efficacy of Intravesical Botulinum Toxin A Injection in the Treatment of Refractory Overactive Bladder in Children"

_jpm, 2023, doi:10.3390/jpm13040616_

Round 1

Reviewer 1 Report

I read with great interest this important study titled, ‘Efficacy of intravesical botulinum toxin A injection in the treatment of refractory overactive bladder in children’. 

I have following comments and suggestions for the authors. 

1.       Does the study sample represent consecutive patients who underwent intravesicular BoNT A injections? Please provide information on how this data was collected, how many patients were initially screened, exclusion criteria, how many patients were excluded and reasons for their exclusion.

2.       Please report any potential sources of bias and any efforts to address these. e.g Were these injections done by only one provider? How many providers were involved? Please report other pharmacological and non-pharmacological interventions done for these patients around the same time as BoNT-A injections and if they could have confounded the results.

3.       Please report if there was any missing data and how that was handled during the statistical analysis.

4.       How was the data for GRA collected?

5.       A big limitation in this study is the lack of control group. An improvement in GRA in both neurogenic and neurogenic bladder patients despite no significant improvement in VUDS parameters (with exception of detrusor pressure in only neurogenic bladder patients) raises the concern that it can be a placebo effect. This possibility can be discussed where the authors discuss this disconcordance between subjective and objective outcome measures. Ultimately, this needs to be evaluated in larger case-controlled studies.

6.       Discuss the generalizability of this study.

7.       I will encourage the authors to reword the conclusion section to give a more cautious overall interpretation of results considering the study limitations.

8.     For Table 1, I suggest to group the urodynamic parameters under subheading of primary and secondary outcomes. 

Author Response

Q1. Does the study sample represent consecutive patients who underwent intravesicular BoNT A injections? Please provide information on how this data was collected, how many patients were initially screened, exclusion criteria, how many patients were excluded and reasons for their exclusion.

Ans: Thank you for your valuable questions. Medical records of consecutive pediatric patients aged up to 18 years who received a first dose of intravesical BoNT-A injection between January 2002 and December 2021 were retrospectively reviewed. (Lines 76-78) All patients underwent videourodynamic studies (VUDSs) before the administration of the first dose of intravesical BoNT-A injection. VUDS were repeated three months after the first injection. Patients with no recorded VUDS after the first injection were excluded from analysis. (Lines 90-93) Nineteen patients were initially screened and four patients were excluded due to no recorded VUDS after the first injection of BoNT-A. (Lines 146-147)

Q2. Please report any potential sources of bias and any efforts to address these. e.g Were these injections done by only one provider? How many providers were involved? Please report other pharmacological and non-pharmacological interventions done for these patients around the same time as BoNT-A injections and if they could have confounded the results.

Ans: Thank you for your valuable suggestions. All the injections were performed by a single surgeon (HCK) (line 97). Because previous medications were failed to treat lower urinary tract symptoms before BoNT-A injection, therefore, no additional pharmacological and non-pharmacological intervention was given after BoNT-A treatment. (Lines 105-108)

Q3. Please report if there was any missing data and how that was handled during the statistical analysis.

Ans: Thank you for your valuable question. There was no any missing data in the present study.

Q4. How was the data for GRA collected?

Ans: Thank you for your valuable question. The children or their parents were asked to rate overall changes in bladder symptoms three months after the first dose of BoNT-A injection. GRA, categorized as -3, -2, -1, 0, 1, 2, and 3, indicated markedly worse, moderately worse, mildly worse, no change, mildly improved, moderately improved, and markedly improved bladder symptoms, respectively. (Lines 126-129)

Q5. A big limitation in this study is the lack of control group. An improvement in GRA in both neurogenic and neurogenic bladder patients despite no significant improvement in VUDS parameters (with exception of detrusor pressure in only neurogenic bladder patients) raises the concern that it can be a placebo effect. This possibility can be discussed where the authors discuss this disconcordance between subjective and objective outcome measures. Ultimately, this needs to be evaluated in larger case-controlled studies.

Ans: Thank you for your valuable suggestions. We have added the contents regarding the discussion of placebo effects on the clinical study of functional urology. (Lines 292-298)

Q6. Discuss the generalizability of this study.

Ans: Thank you for your valuable suggestions. We have added the contents regarding the generalizability of this study. All the participants were Taiwanese and recruited only from Hualien Tzu Chi Hospital. The results may not be generalizable outside of this context for other ethnic groups and clinics administering BoNT-A treatment where the procedures of BoNT-A injections might be different. (Lines 311-314)

Q7. I will encourage the authors to reword the conclusion section to give a more cautious overall interpretation of results considering the study limitations.

Ans: Thank you for your valuable suggestions. We have revised the conclusion section. (Lines 335-337)

Q8. For Table 1, I suggest to group the urodynamic parameters under subheading of primary and secondary outcomes. 

Ans: Thank you for your valuable suggestion. We have divided the statements of primary and secondary outcome into different paragraphs with individual and subheading. (Line 110 and line 120)  Primary outcome measures of the present study were differences in VUDS parameter values regarding bladder storage function before and after the first dose of intravesical BoNT-A injection, including compliance, detrusor pressure and cystometric capacity. The secondary outcome measure was determined by the GRA measured three months after the first dose of BoNT-A injection. Accordingly, all the urodynamic parameters in table 1, excluding post-void residual (not the outcome measure in the present study), were primary outcomes. (Lines 111-123)

Reviewer 2 Report

A well thought out paper. I appreciate the opportunity to review this study.The study results have demonstrated that intravesical BTX-A injection is effective and safe for the treatment of neurogenic and non-neurogenic OAB in children .

I just have few comments.

1- How did you define neurogenic OAB ? Three patients with spinal cord injury participanted in the study?Please added the patients related information such as the level of neurological injury,the American Social Injury Association grade(AIS),duration of spinal cord injury .

2- Did fifteen pediatric patients received repeated injections?Repeated intravesical BTX-A injections seem to be possible without loss of efficacy ?

Author Response

Q1. How did you define neurogenic OAB ? Three patients with spinal cord injury participated in the study?Please added the patients related information such as the level of neurological injury,the American Spinal Injury Association grade(AIS),duration of spinal cord injury .

Ans: Thank you for your valuable questions. Neurogenic OAB was defined as OAB occurs as a result of an injury or disorder of any part of the nervous system, including spinal cord injury, spinal dysraphism, or intracranial lesions. (Please refer to lines 136-138)

Two patients with spinal cord injury participated in the study. In two patients with spinal cord injury, one was diagnosed as T7-11 spinal cord injury with AIS grade A one year prior to the first dose of BoNT-A, and the other one was diagnosed as C4-5 spinal cord injury with AIS grade A two years before the first dose of BoNT-A. (Lines 150-154)

Q2. Did fifteen pediatric patients received repeated injections?Repeated intravesical BTX-A injections seem to be possible without loss of efficacy?

Ans: Thank you for your valuable questions. Only 12 patients in the present study received repeated injections. Because the case number of pediatric patients underwent repeated BoNT-A injections decreased further and the regularity of follow-up was not optimal, we did not analyze the efficacy of repeated BoNT-A injections.

Round 2

Reviewer 1 Report

The authors have adequately addressed the concerns raised in the initial report.